# Current khat (*Catha edulis* F.) use among Ethiopian women and its association with anemia and underweight: A cross-sectional analysis from Ethiopian Demographic and Health Survey

**Nebyu Daniel Amaha**[1]*, **Meron Mehari Kifle**[2], **Samson Goitom Mebrahtu**[3]

1 Department of Nutrition and Dietetics, College of Health Sciences, Mekelle University, Tigray, Ethiopia,
2 Nuffield Department of Clinical Medicine, Centre for Tropical Medicine and Global Health, University of Oxford, Oxford, United Kingdom, 3 Kiel University, Kiel, Germany

* nebyudan@gmail.com

**Data Availability Statement:** All relevant data are within the paper and its Supporting Information files. Also, the stata code data underlying the

## Abstract

### Background

Chewing fresh leaves of *Catha edulis* (khat) is a popular pastime activity among Ethiopians where 12% women chew it. Reports show that khat use has been associated with poor nutritional status. This study aimed to determine whether khat chewing is linked to underweight and anemia.

### Method

We analyzed data from the 2016 Ethiopian Demographic and Health Survey (EDHS). The EDHS used two stage stratified cluster sampling to collected data from 16,650 households. We used data from a total of 15,683 respondents and 1904 respondents who chewed ever chewed khat in their lives. We used Pearson's chi-square, and logistic regression while stratifying by residence (urban vs rural) to control for confounders.

### Result

Our results indicated 10.7% (95%CI: 10.92,11.26) of women chewed khat for an average of 16.5 days in the previous month. A woman's current khat chewing status was significantly associated with age, educational level, region, religion, wealth group, and marital status. Women aged 40–44 were significantly more likely to chew (AOR = 2.89,) compared to those aged 15–19. Compared with Protestant women, Muslim women were 210 times more likely (AOR = 210, 95% CI 102,435.7) to chew and women in the poorest wealth quintile had 73% higher odds (AOR = 1.73, 95% CI 1.22,2.44) of chewing khat when compared with the richest. Anemia was not associated with a woman's chewing status, whereas rural women who chewed khat for more than 26 days in a month had a 78% increased risk (OR = 1.78) of being underweight when compared to non-chewers.

results presented in the study are available from https://github.com/NebyuDanielAmaha/ EDHSAnalysisSTATA/blob/main/EDHS_Stata_ Analysis.do

**Funding:** The author(s) received no specific funding for this work.

**Competing interests:** The authors have declared that no competing interests exist.

## Conclusion

Khat chewing is associated with sociodemographic factors and current khat use is associated with a higher risk of underweight among women living in rural areas. Implementing targeted awareness campaigns for women about the risks of khat chewing is recommended.

## Introduction

Khat (*Catha edulis* Forskal) is an indigenous plant to eastern Africa, known for its stimulant properties due to the presence of alkaloids cathinone and cathine, which are structurally similar to amphetamine [1,2]. The consumption of khat, typically through chewing, results in the absorption of these alkaloids into the bloodstream, leading to a range of physiological effects such as increased alertness, energy, mood elevation, social disinhibition, appetite suppression, and fatigue reduction [3,4]. However, the habitual use of khat has been linked to various physiological and psychological disorders, including insomnia, anorexia, hypertension, increased risk of acute myocardial infarction, gastritis, hemorrhoids, constipation, duodenal ulcers, and gastroesophageal reflux [3,5–7].

The practice of khat chewing, often incorporated into traditional and religious ceremonies, is a prevalent social activity in several regions globally, particularly in Ethiopia. The Demographic and Health Survey [8] reported a prevalence of khat chewing among Ethiopian women at approximately 12%. Another nationally representative survey conducted in 2015, the STEPwise approach to NCD risk factor surveillance (STEPS), found a lifetime prevalence of 9.4% among Ethiopian women [9]. A study conducted among Yemeni women in the city of San'a reported that 29.6% of women chewed khat [4]. Interestingly, the habit of khat chewing among women does not appear to decrease during pregnancy or breastfeeding. Several cross-sectional studies have reported the prevalence of khat chewing during pregnancy to be around 15% in Eastern Ethiopia [10], 11% in Southern Ethiopia [11], 27.4% in South West Ethiopia [12], and 40.7% in Yemen [2]. This is of particular concern given the potential negative health implications of khat chewing, including malnutrition [13–15].

Malnutrition remains a significant public health issue in Ethiopia, with 22% of Ethiopian women reported as underweight (BMI < 18.5) and 24% as anemic (Hemoglobin <11g/deciliter) (EDHS). While there is a growing body of research investigating the health effects of khat chewing in Ethiopia, the majority of these studies are small-scale, cross-sectional studies conducted in specific geographical areas, and the findings are often conflicting. Some studies have reported an association between khat chewing and increased odds of malnutrition among women in Addis Ababa [13]. For instance, a study conducted in a subcity of Addis Ababa found that khat-chewing women were 2.1 times more likely to be underweight than non-khat-chewing women [13]. Another study in Eastern Ethiopia reported that khat-chewing women had a 29% higher risk of being anemic than non-khat-chewing women [14]. However, other studies have reported contradictory findings. For example, Tadele et al. found that women who chew khat had higher BMI and weight than non-chewing women [16], and an analysis of factors associated with underweight in women found that women who chew khat had lower odds of being underweight [17]. These conflicting findings underscore the need for further research to explore the association between khat chewing and malnutrition in women.

The primary objective of this study is to to investigate the association between khat chewing and malnutrition. This study represents the first analysis to investigate the association between khat chewing and malnutrition using a nationally representative data set. Previous analyses of

demographic health survey studies [18,19] have primarily focused on factors associated with khat chewing and lifetime prevalence of khat chewing, rather than current khat chewing status. It is crucial to acknowledge that a woman who previously engaged in khat chewing may not necessarily be actively chewing khat at the time of the interview. And because we're trying to measure whether khat chewing is associated with malnutrition, we included women who were actively engaged (defined as chewing khat at least once in the previous 30 days) during the study period.

## Methods

### Study design and sampling

This study used a secondary data analysis methodology, utilizing information extracted from the 2016 Ethiopian Demographic and Health Survey (EDHS). The EDHS is a nationally representative sample that provides estimates at the national, regional, urban, and rural levels. The sample was stratified and selected in two stages from the 84,915 enumeration areas based on the 2007 national census. Detailed demographic and health-related information was collected from children living in 16,650 households, 15,683 female respondents, and 12,688 male respondents aged 15–59 using five questionnaires [8]. The dataset is freely available upon reasonable request and can be accessed from the DHS website [20].

The study sample included all females aged 15–49. The primary independent variable for this study was the khat chewing status of a woman, obtained from the "Woman's Questionnaire" of the EDHS 2016. The variables of interest were questions number 1107a and 1107b. The first question asked, "Have you ever chewed khat?" with a yes or no option. The second question asked, "During the last 30 days, how many days did you chew khat?". The selection of the final sample is shown in the figure below (Fig 1). For the final analysis, we included women who chewed khat at least once in the last 30 days prior to the survey. We assumed that because khat chewing is a highly addictive habit, a woman who did not chew in the last month was less likely to be considered a current chewer.

### Study variables and measurements

The outcomes of interest were current anemia, measured in hemoglobin (Hg) level of anemic (Hg < 11g/dL), and underweight (measured as BMI < 18.5). The independent variables were khat use among women and its frequency. Previous studies found that age, region, educational level, employment, marital status, religion, wealth index, and residence were factors associated with khat chewing among women [9,10,19,21]. Weight measurements were carried out using lightweight SECA® scales with a digital screen designed and manufactured under the guidance of UNICEF. Height measurements were carried out using a Shorr® measuring board. The EDHS 2016 collected blood specimens for anemia testing from women aged 15–49 who voluntarily consented. Hemoglobin analysis was carried out on-site using a battery-operated portable HemoCue ® analyzer [8].

### Data analysis

Descriptive statistical analyses were used to determine the current prevalence of khat chewing behavior among Ethiopian women aged 15–49. Pearson's chi-square test and multivariate logistic regression were carried out to analyze the association between socio-demographic characteristics and current khat chewing practice and between number of khat chewing days and malnutrition. As recommended by the DHS Measure program, we used sample weight (V005/1000000) to compensate for

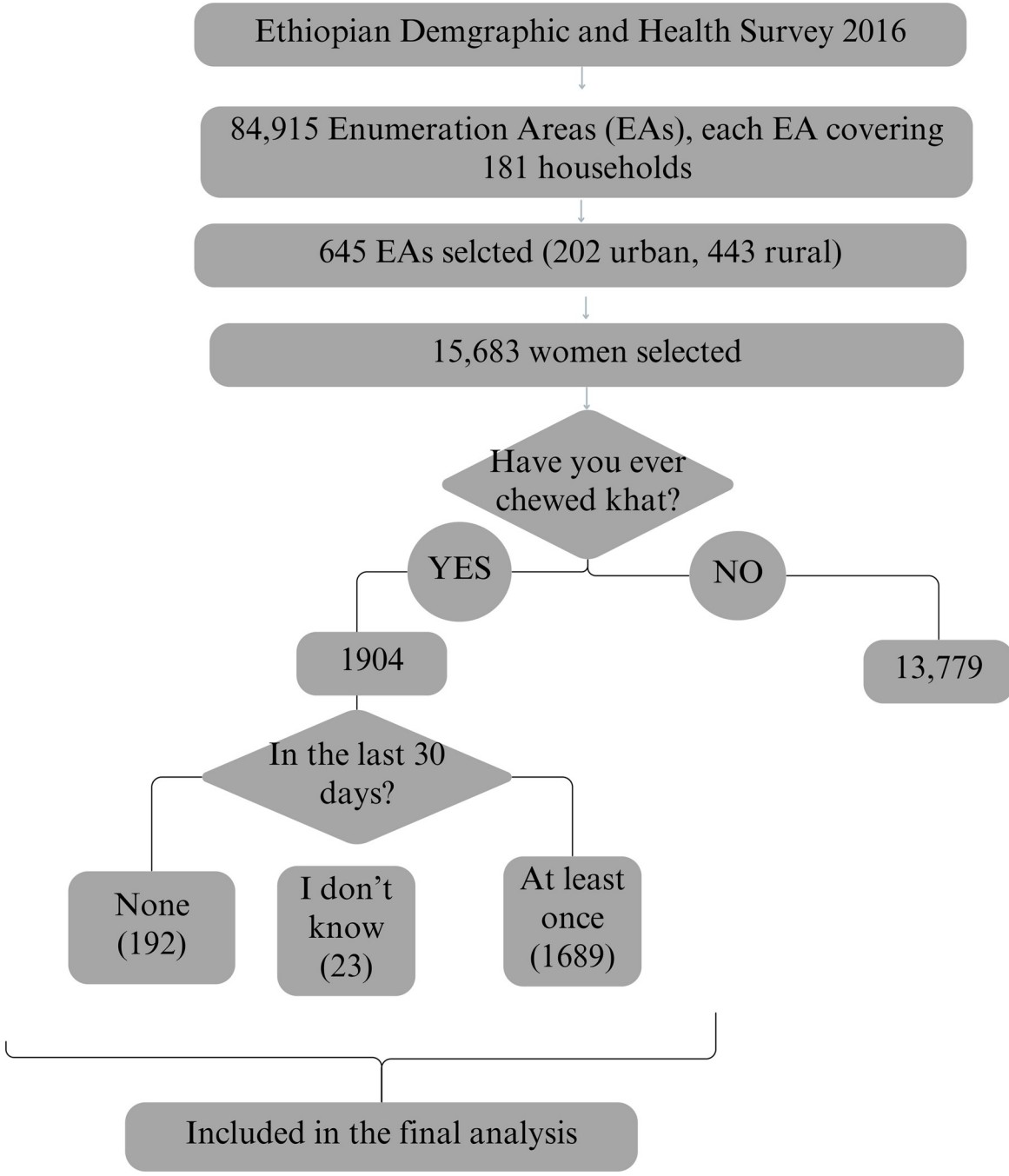

**Fig 1. Sample selection process to identify eligible women to be included in the study to determine the association between khat chewing and malnutrition.**

the unequal probability of selection used during study design and data collection [22]. We used STATA version 13 for the statistical analysis. The do code for the STATA code is available in the following GitHub repository [23].

## Results section

### Socio-demographic profile of all participants

Our analysis found that 10.7% (95%CI:10.92,11.26) of participants reported chewing khat in the past month. The majority of participants resided in rural areas (77.8%) and belonged to the younger age group (15–19 years old, 21.6%). Additionally, nearly half lacked formal education (47.8%), with a significant portion from the Oromia region (36.4%) and identifying as Orthodox Christian (43.3%). Notably, the wealthiest wealth index category had the highest prevalence (26.5%) of khat chewing. Interestingly, most women were not employed (66.7%) and were either married (63.9%) or never married (25.7%). Maternal status breakdown showed 30.7% breastfeeding and 7.2% pregnant. These findings suggest potential links between khat chewing and sociodemographic factors, particularly residence, age, education, and employment. The high prevalence among wealthier individuals and potential impact on maternal health warrant further investigation (Table 1).

Our analysis showed that 10.7% of women chewed khat at least once in the last 30 days. Demographically, the majority of the respondents resided in rural areas (77.8%), fell within the 15–19 age category (21.6%), lacked formal education (47.8%), lived in the Oromia region (36.4%), adhered to the Orthodox Christian faith (43.3%), and belonged to the wealthiest wealth index (26.5%). Approximately two-thirds of respondents were not employed (66.7%) at the time of the interview. Marital status distribution indicated that the majority were either married (63.9%) or had never been married (25.7%). In terms of maternal status, 30.7% of respondents were breastfeeding, and 7.2% were pregnant (Table 1).

### Association between khat chewing frequency and malnutrition

Our analysis found that 23.1% and 24.8% of Ethiopian women surveyd were underweight and anemic, respectively (Table 2). According to Pearson's chi-square test of association, there is a significant association between BMI and current khat chewing (chi-square = 40.6, df = 4, p<0.001), while no association was observed between anemia and khat chewing status (chi-square = 40.6, df = 3, p = 0.139) (S1 Table). The results showed that 40.8% of women chewed 26–30 days in the last 30 days whereas 11.3% didn't chew khat in the last month. Further analysis showed that 39.2% of women chewed on a daily basis and the average number of khat chewing days was 16.46 days (95%CI: 15.89–17.02) (Table 2). Similar to the aggregated association between BMI and current khat chewing status, number of days a woman chewed khat is associated with being underweight (chi-square:17.7,df = 6,p = 0.007) but not with anemia (chi-square = 8.4,df = 6,p = 0.211) (Table 2).

The mean BMI of women who did not chew khat in the previous month of the study (0 days) was higher than that of women who engaged in khat chewing. Chewing khat for fewer than 5 days does not seem to impact the odds of being underweight (OR = 1.48, p = 0.12). However, after chewing for6 days and more, the odds of being underweight increased significantly. Women who chew khat almost daily (26–30 times/month) were 2.54 times more likely to be underweight than those who did not chew khat. The highest risk of underweight appears to be among women who chew khat 16–20 days (OR = 3.1, p = 0.0007). Chewing khat for 1 to 5 days in a month was not significantly associated with underweight (p = 0.1277) (Table 3).

To assess potential differences in BMI and hemoglobin levels based on current khat chewing status, an independent t-test was conducted on an unweighted sample of 1459 women. This sample comprised 246 non-chewing individuals and 1213 women who were currently chewing in the previous 30 days. The findings revealed that participants in the chewing group exhibited statistically significantly lower BMI (21.5 ± 4.14 units) compared to those in the

**Table 1. Socio-demographic characteristics of the women participants in 2016 Ethiopian Demographic and Health Survey (EDHS) and its association with their current khat chewing status (N = 15,683).**

| | Variables | Sample | | Currently chewing | | | |
|---|---|---|---|---|---|---|---|
| | | N | % | Yes (n)[a] | % | $X^{2b}$ | p-value |
| Age category in years | 15–19 | 3,381 | 21.6 | 224.38 | 6.64 | 220.18 | < 0.001 |
| | 20–24 | 2,762 | 17.6 | 244.23 | 8.84 | | |
| | 25–29 | 2,957 | 18.9 | 354.62 | 11.99 | | |
| | 30–34 | 2,345 | 15.0 | 314.11 | 13.39 | | |
| | 35–39 | 1,932 | 12.3 | 239.06 | 12.37 | | |
| | 40–44 | 1,290 | 8.2 | 171.74 | 13.32 | | |
| | 45–49 | 1,017 | 6.5 | 140.83 | 13.85 | | |
| Type of place of residence | Urban | 3,476 | 22.2 | 224.53 | 6.46 | 1.7148 | 0.190 |
| | Rural | 12,207 | 77.8 | 1,464.46 | 12.00 | | |
| Highest educational level | No education | 7,498 | 47.8 | 1,074.37 | 14.33 | 87.844 | <0.001 |
| | Primary | 5,490 | 35.0 | 531.541 | 9.68 | | |
| | Secondary | 1,817 | 11.6 | 64.33 | 3.54 | | |
| | Higher | 877 | 5.6 | 18.75 | 2.14 | | |
| Region | Tigray | 1,129 | 7.2 | 5.26 | 0.47 | 1800 | <0.001 |
| | Afar | 128 | 0.8 | 8.019 | 6.26 | | |
| | Amhara | 3,714 | 23.7 | 211.206 | 5.69 | | |
| | Oromia | 5,701 | 36.4 | 1,283.34 | 22.51 | | |
| | Somali | 459 | 2.9 | 10.130 | 2.20 | | |
| | Benishangul | 160 | 1.0 | 4.4360 | 2.77 | | |
| | SNNP‡ | 3,288 | 21.0 | 107.817 | 3.28 | | |
| | Gambela | 44 | 0.3 | 1.298 | 2.97 | | |
| | Harari | 38 | 0.2 | 11.759 | 30.56 | | |
| | Addis Adaba | 930 | 5.9 | 23.2973 | 2.50 | | |
| | Dire dawa | 90 | 0.6 | 22.435 | 24.82 | | |
| Religion | Orthodox | 6,786 | 43.3 | 93.00 | 1.37 | 1300 | <0.001 |
| | Catholic | 120 | 0.8 | .0965 | 0.08 | | |
| | Protestant | 3,674 | 23.4 | 7.5022 | 0.20 | | |
| | Muslim | 4,893 | 31.2 | 1,578.42 | 32.26 | | |
| | Traditional | 123 | 0.8 | 9.975 | 8.10 | | |
| | Other | 87 | 0.6 | 0 | 0.00 | | |
| Wealth index combined | Poorest | 2,633 | 16.8 | 329.003 | 12.50 | 73.17 | <0.001 |
| | Poorer | 2,809 | 17.9 | 463.921 | 16.51 | | |
| | Middle | 2,978 | 19.0 | 398.204 | 13.37 | | |
| | Richer | 3,100 | 19.8 | 240.892 | 7.77 | | |
| | Richest | 4,163 | 26.5 | 256.982 | 6.17 | | |
| Currently working | No | 10,463 | 66.7 | 1,259.02 | 12.03 | 0.0005 | 0.983 |
| | Yes | 5,220 | 33.3 | 429.977 | 8.24 | | |
| Current marital status | Never in union | 4037 | 25.7 | 184.918 | 4.58 | 208.84 | <0.001 |
| | Married | 10,014 | 63.9 | 1,337.76 | 13.36 | | |
| | Living with partner | 209 | 1.3 | 24.032 | 11.50 | | |
| | Widowed | 429 | 2.7 | 37.678 | 8.78 | | |
| | Divorced | 764 | 4.9 | 89.5366 | 11.73 | | |
| | Separated | 230 | 1.5 | 15.0740 | 6.54 | | |
| Currently pregnant | No or unsure | 14,548 | 92.8 | 1,523.80 | 10.47 | 3.8156 | 0.051 |
| | Yes | 1,135 | 7.2 | 165.202 | 14.55 | | |

*(Continued)*

**Table 1.** (Continued)

| | Variables | Sample | | Currently chewing | | | |
|---|---|---|---|---|---|---|---|
| | | N | % | Yes (n)[a] | % | X$^{2b}$ | p-value |
| Currently breastfeeding | No | 10,870 | 69.3 | 1,067.62 | 9.82 | 0.945 | 0.331 |
| | Yes | 4,813 | 30.7 | 621.377 | 12.91 | | |
| Total | | 15,683 | | 1689 | 10.77 | | |

SNNP, South Nations and Nationalities region

[a]the fractions of sample sizes are because the sample is weighted

[b]chi-square.

non-chewing group (22.9 ± 4.47 units), t(1457) = 4.82, p< 0.0001. However, there was no significant difference in hemoglobin levels between the two groups (p = 0.78).

To address the confounding effects of residence, the test of association between number of khat chewing days and underweight was carried out by stratifying for residence. The association between number of days and underweight showed mixed results for women living in urban areas. However, lower than average BMI was associated with women living in the rural areas if a woman chewed khat for 26 to 30 days. A rural woman was 78% more likely (AOR = 1.78, 95%CI: 1.05–3.02, p = 0.031) to be underweight if she chewed khat for 26–30 days when compared with those who didn't chew khat. No significant association was observed for women who chewed khat for less than 26 days (Table 4).

## Discussion

The study aimed to identify the association between khat chewing and malnutrition, specifically underweight and anemia. Our results showed that, female khat chewers showed a significant association with being underweight, but not with anemia. The analysis revealed connections with age, region, religion, wealth index, and marital status, while educational level, residential area, and employment status did not show noteworthy associations. Additionally, the frequency of khat chewing in women was linked to being underweight.

Our initial analysis revealed a significant correlation between khat chewing and maternal BMI (Table 4). However, previous study revealed that women who lived in rural areas are 2.25

**Table 2. Association between the number of khat chewing days with underweight and anemia among current women chewers using Pearson's chi-square test (N = 1904).**

| Days | n (%) | Underweight[a] | | Chi-2[c] | p-value | Anemic[b] | | Chi-2 | p-value |
|---|---|---|---|---|---|---|---|---|---|
| | | Yes (%) | No | | | Yes (%) | No | | |
| 0 | 215 (11.3) | 23 (12.7) | 158 | 17.76 | 0.007 | 37 (18.2) | 166 | 8.38 | 0.211 |
| 1–5 | 459 (24.1) | 72 (17.8) | 332 | | | 104 (23.1) | 345 | | |
| 6–10 | 141 (7.4) | 33 (28.3) | 85 | | | 36 (28.2) | 92 | | |
| 11–15 | 165 (8.7) | 29 (23.3) | 97 | | | 38 (23.9) | 123 | | |
| 16–20 | 86 (4.5) | 24 (31.1) | 53 | | | 25 (30.5) | 56 | | |
| 21–25 | 62 (3.2) | 14 (27.7) | 36 | | | 20 (34.1) | 40 | | |
| 26–30 | 777 (40.8) | 171 (27.0) | 461 | | | 189 (26.0) | 537 | | |
| Total | 1904 (100) | 366 (23.1) | 1221 | | | 449 (24.8) | 1358 | | |

[a]defined by BMI< 18.5

[b]defined as Hg<11g/dL

[c]:chi-square.

**Table 3. Logistic regression showing association between number of days a woman chewed khat and malnutrition along with the mean BMI and hemoglobin levels for the day categories.**

| Days | Mean BMI | Underweight[a] | | | Mean Hg g/dL | Anemia[b] | | |
|---|---|---|---|---|---|---|---|---|
| | | COR | 95%CI | p-value | | COR | 95%CI | p-value |
| 0 | 22.93 | Ref | - | - | 13.66 | Ref | - | - |
| 1–5 | 22.25 | 1.48 | 0.89, 2.46 | 0.1277 | 13.29 | 1.36 | 0.89, 2.06 | 0.1500 |
| 6–10 | 21.31 | 2.70 | 1.49, 4.89 | 0.0010 | 13.13 | 1.77 | 1.05, 2.99 | 0.0328 |
| 11–15 | 21.57 | 2.09 | 1.14, 3.81 | 0.0164 | 13.07 | 1.41 | 0.85, 2.35 | 0.1813 |
| 16–20 | 21.23 | 3.10 | 1.62, 5.96 | 0.0007 | 13.29 | 1.98 | 1.09, 3.59 | 0.0243 |
| 21–25 | 19.82 | 2.63 | 1.23, 5.62 | 0.0127 | 13.28 | 2.33 | 1.23, 4.43 | 0.0098 |
| 26–30 | 21.06 | 2.54 | 1.58, 4.07 | 0.0001 | 13.12 | 1.58 | 1.07, 2.34 | 0.0224 |

COR: Crude Odds Ratio, BMI: Body Mass Index

[a]defined by BMI< 18.5

[b]defined as Hg<11g/dL.

more likely to be underweight when compared with their urban counterparts (17). When we analyzed the association by controlling for residence, we found out that number of days a woman chewed khat showed different results between urban and rural women. The association of underweight with number of days a woman chewed khat among the urban women khat showed that the number of days is mixed showing that there are other factors associated with underweight apart from khat. Among the rural women, the association seems clearer. No significant association was observed between being underweight and 25 days of khat chewing in the last 30 days. The only significant association is when a woman chews khat for more than 26 days or more. Similar to our findings, Kassie et al. reported khat chewing does not lead to higher odds of being underweight, in fact non-chewers were 51% (AOR = 1.51, p = 0.040) more likely to be underweight when compared with their khat chewing counterparts (17). This disparity could arise from the study considering the lifetime prevalence of khat rather than the current khat chewing status of women. Another analysis of the STEPs survey did not identify any significant difference between khat chewers and non-chewers of both sexes (16). However, there are studies, alebit smaller in size, which report significant association between lower BMI and khat chewing from Jimma town, Ethiopia (15) and Addis Ababa, (13) which indicate that khat chewers were more likely to be underweight than non-khat chewers. Although we did not find a direct association between khat chewing and anemia, there are reports indicating an association between khat chewing and anemia among pregnant women (10, 14). This

**Table 4. Logistic regression stratified by residence of khat chewing women showing association between the number of khat chewing days and odds of being underweight (<18.5 BMI) (N = 1904).**

| Days | Odds of underweight | | | | | |
|---|---|---|---|---|---|---|
| | Urban | | | Rural | | |
| 0 | Ref | 95%CI | p-value | Ref | | p-value |
| 1–5 | 1.34 | 0.35,5.05 | 0.6683 | 1.15 | 0.65,2.03 | 0.62 |
| 6–10 | 38.68 | 8.93,167.35 | <0.0001 | 1.24 | 0.62,2.45 | 0.531 |
| 11–15 | 4.67 | 1.07,20.26 | 0.0397 | 1.38 | 0.70,2.69 | 0.347 |
| 16–20 | 14.99 | 2.58,86.89 | 0.0025 | 1.83 | 0.89,3.74 | 0.0993 |
| 21–25 | 11.68 | 1.93,70.38 | 0.0073 | 1.51 | 0.64,3.53 | 0.3401 |
| 26–30 | 0.48 | 0.06,3.67 | 0.4779 | 1.78 | 1.05,3.02 | 0.0314 |

association during pregnancy may be attributed to increased nutritional demands rather than khat chewing itself.

When examining the mean BMI of women who chewed khat based on the number of days, a notable association emerged (Chi-square = 17.76, p = 0.007) (Table 3). For instance, women who did not chew khat had a mean BMI of 22.93, whereas those who chewed almost daily had a mean BMI of 21.06 (Table 3). Women who chewed daily had 78% higher odds of being underweight compared to non-chewers among the rural women. Chewing khat 1 to 25 days in the previous month did not result in significantly higher odds of being underweight. This suggests that a woman needs to chew khat almost daily in a month to have higher odds of being underweight. When analyzing the 2016 EDHS data for factors associated with underweight, Kassie et al found that women in the rural areas were twice more likely (AOR = 2.25 95% CI:1.57–3.23 p<0.001) to be underweight than her urban counterpart. Thus we analyzed the association between number of khat chewing days and underweight by residence (urban vs rural). The stratified results showed that khat chewing was mostly not associated with being underweight in rural areas unless the woman chewed for more than 26 days in the last month when compared with those who didn't chew (Table 4). The results from the urban strata is inconclusive with the highest and lowest age category days of khat chewing showing no significant association, whereas, the rest of the categories showing varying degrees of association. Therefore, khat chewing and its association with number of days seems to be more pronounced among women living in rural areas. Comprehensive qualitative research is essential to unpack the cultural, economic, and lifestyle factors underpinning khat chewing in Ethiopia. Deep dives through interviews, ethnography, and engagement in relevant rituals can illuminate the lived experiences and perspectives of khat using among women. This knowledge will inform culturally sensitive approaches to khat's challenges and opportunities.

## Strengths and limitations

This study utilized a nationally representative sample to estimate the current prevalence of khat chewing among Ethiopian women aged 15–49, allowing for the generalization of findings to the target population. The research was conducted by proficient professionals using standardized methods, thereby minimizing the potential for measurement errors. However, there are certain limitations to our analysis. Although we accounted for the frequency of khat chewing, we did not consider factors such as the dosage of khat chewed per session and the duration of a woman's khat chewing habit. Despite these limitations, this study stands as the first investigation of its kind to examine the association between khat chewing and malnutrition among women at a national level. The findings contribute to bringing much-needed attention to the issue of khat chewing among women. Future studies should include the number of years a woman has been chewing khat and the dosage of khat consumed per session.

## Conclusion

The present study reveals that 10.7% of Ethiopian women engage in khat chewing, a prevalence seemingly higher than reported in the previous EDHS. Adjusted logistic regression analysis identified age, region, and religion as the primary factors associated with a woman's khat chewing status in Ethiopia. Additionally, women who chewed khat were found to be more likely to being underweight than non-chewers. Furthermore, the number of days a woman spent chewing khat had a direct impact on the risk of being underweight. Those who chewed for less than 6 days did not exhibit a significantly higher risk of low BMI compared to non-chewers. Consequently, there is a need for policy and educational interventions aimed at

raising awareness about the health effects of khat, especially in Muslim-majority regions of Ethiopia.

## Supporting information

**S1 Table. Bivariate and multivariate analysis showing association between current khat chewing status and malnutrition using Pearson's chi-square.**
(DOCX)

## Acknowledgments

The authors would like to thank the DHS Program for allowing us to use their 2016 Ethiopian Demographic and Health Survey dataset.

## Author Contributions

**Conceptualization:** Nebyu Daniel Amaha.

**Data curation:** Nebyu Daniel Amaha.

**Formal analysis:** Nebyu Daniel Amaha, Meron Mehari Kifle.

**Investigation:** Nebyu Daniel Amaha.

**Methodology:** Nebyu Daniel Amaha, Meron Mehari Kifle.

**Software:** Nebyu Daniel Amaha.

**Validation:** Nebyu Daniel Amaha, Meron Mehari Kifle.

**Visualization:** Nebyu Daniel Amaha.

**Writing – original draft:** Nebyu Daniel Amaha, Samson Goitom Mebrahtu.

**Writing – review & editing:** Nebyu Daniel Amaha, Samson Goitom Mebrahtu.

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
