## [Decision Letter · Decision Letter 0]

22 Mar 2024

PONE-D-24-01395Prevalence and correlates of khat (Catha edulis F.) use, anemia, and underweight in Ethiopian women: A cross-sectional analysis from Ethiopian Demographic and Health SurveyPLOS ONE

Dear Dr. Amaha,

Thank you for submitting your manuscript to PLOS ONE. After careful consideration, we feel that it has merit but does not fully meet PLOS ONE’s publication criteria as it currently stands. Therefore, we invite you to submit a revised version of the manuscript that addresses the points raised during the review process.

**Please address all the reviewers comments and edit the language of the entire manuscript by native English language speaker.**

We look forward to receiving your revised manuscript.

Kind regards,

Melese Shenkut Abebe, PhD

Academic Editor

PLOS ONE

Reviewers' comments:

Reviewer's Responses to Questions

**Comments to the Author**

1. Is the manuscript technically sound, and do the data support the conclusions?

Reviewer #1: Yes

Reviewer #2: No

2. Has the statistical analysis been performed appropriately and rigorously? 

Reviewer #1: Yes

Reviewer #2: No

3. Have the authors made all data underlying the findings in their manuscript fully available?

Reviewer #1: Yes

Reviewer #2: Yes

4. Is the manuscript presented in an intelligible fashion and written in standard English?

Reviewer #1: Yes

Reviewer #2: No

5. Review Comments to the Author

Reviewer #1: The manuscript is well written generally. However, there are minor comments to be addressed.

1. There is a need to revise the whole manuscript for typographical errors.

2. In the methodology part the authors stated an inclusion criteria as women who chewed khat at least once in the last 30 days. what is your background to make this 30 days as a cut of point? any reference??

3.in the discussion art the authors tried to explain the higher odd of chewing khat at older age might be due to higher financial independence yet a few lines after, there is a contradictory explanation stating , unemployed individuals are more likely to chew khat. How can you explain these two assumptions?

4. Do the authors have any methodological recommendation for further study as the current study mentioned the limitations to consider factors such as the dosage of khat chewed per session and the duration of a woman's?

Reviewer #2: Dear!!! you have to do again you report to make different from previous study on Prevalence and determinants of chewing khat among women in Ethiopia: data from Ethiopian Demographic and Health Survey 2016

6. PLOS authors have the option to publish the peer review history of their article (what does this mean?). If published, this will include your full peer review and any attached files.

Reviewer #1: **Yes: **Zelalem Animaw

Reviewer #2: No

---

## [Author Response · Author response to Decision Letter 0]

9 Apr 2024

1. From title to conclusion

a) The title, the abstract, the objective, and the result are not in line

Response: The title, abstract and objective have been updated to come in line with the results. After considering the comments, we have focused the title, objective to focus on the association between khat chewing and malnutrition. We left out factors associated with khat chewing in the objective, results and discussion.

b) Lack of concentration or focus according to the topics

Response: The manuscript been improved to give enough emphasis to the revised objective.

c) Poor language use

Response: The language use has been revised based on the reviewers comments.

d) Do you think your study is different from Prevalence and determinants of chewing khat among women in Ethiopia: data from Ethiopian Demographic and Health Survey 2016

Response: Yes it is different because it investigates the association between khat use among Ethiopian women and malnutrition. Secondly, this study considered the determinant of khat use on current use not life time use of khat.

e) The same study with Yimenu Yitayih & Jim van Os study above but your report is different from the previous one. Why?

Response: Our report is different on two main points. First because it analyzes the relationship between khat chewing and its association with malnutrition whereas Yitayeh and van OS studied only the factors associated with khat chewing. Second, we studied the factors associated with current (most recent) khat chewing behavior and not life time khat chewing. 

2. Abstract

a) Not structure

Response: The abstracted has been converted into a structured format which includes background, method, result, and conclusion.

b) Lacking a study population, sampling, and data collection

Response: The abstract has has been edited to include the above points.

c) Lacking the absolute number of participants, the result of prevalence and its CI

Response: The absolute number of participants 15,683 women has been added. Moreover, the sampling and data collection has been mentioned under the methods section of the abstract.

d) The conclusion without recommendation

Response: Recommendation has been added.

e) No keyword

Response: The following keywords have been added: khat, malnutrition, anemia, underweight 

3. Introduction

a) Line 60-61: What is STEEP

Response: The full meaning of STEPS is given in the same line. It means (STEPwise approach to NCD risk factor Surveillance)

b) Line 85-88: Long sentence

Response: The long sentence has been modified to include the main objectives of the study.

4. Method

a) Poor structure

Response: The structure of the method section has been modified based on the comment of the reviewers. It now reads as =>Study design and sampling; =>study variables and Measurements =>Data analysis

b) 115-126: Unclear information, Why did you report other studies here? Is it a case-control study?

Response: The section reporting other studies has been deleted.

c) Line 131 and 139: Measurements

Response: The paragraph has been moved under the study sample and measurements sub section of the methodology.

5. Result

a) Line 155-161:Poor summary of Socio-demographic profile

Response: The summary has been improved for clarity.

b) Line 155: Prevalence without Confidence internal. It is not part of the Socio-demographic profile

Response: The confidence interval has been added for the prevalence of khat chewing women (95%CI:10.92-11.26). It is part of the socio-demographic profile as can be seen on Table 1, last row titled “Total”.

c) 158: adhered to the Orthodox Christian faith. What do you mean?

Response: This sentence has been improved to “…identifying as Orthodox Christian..”

d) Line: 170-185 Rewrite it with clear statements Socio-demographic characteristics of current khat chewers

Response: The whole paragraph has been rewritten with more coherent and clearer style.

e) No report on the prevalence of malnutrition

Response: A report on underweight and anemia has been added based on the results from Table 2.

f) Line 212: Association between khat chewing frequency and malnutrition. Why do we use only age as a predictor variable?

Response: The number of days a woman chewed khat is used as a predictor of malnutrition based on the assumption that if a woman chews khat very often it’ll likely increase in higher intake of khat and the association with malnutrition would be higher than a woman who chews khat less frequently.

6. Discussion

a) It is not concise because of the title objective and result

Response: The section of the discussion “prevalence of current khat chewing” has been removed to bring the discussion in line with the objectives. The overall style of the discussion has been revised to avoid verboseness.

7. Conclusion and recommendation

a) Line 431-432: I did not see it in the regression, but you reported it as an associated factor.

Response: It has been removed because it wasn’t included in the results section.

b) Line 434-435: women who chewed khat were found to be more susceptible to being underweight than non-chewers. How did measure susceptibility? Where is it in the results?

Response: The work susceptible has been replaced with likely to avoid confusion. It is referring to the higher odds of being underweight.

8. There is a need to revise the whole manuscript for typographical errors.

Response: The manuscript has been modified to follow the PLOS One formatting guidelines.

9. In the methodology part the authors stated an inclusion criteria as women who chewed khat at least once in the last 30 days. What is your background to make this 30 days as a cut of point? any reference??

Response: The cut point is based on the questionnaire conducted by the Ethiopian Demographic and Health Survey. Based on the second question, “how many days in the last 30 did you chew khat?” we decided to take this cut point to refer to a woman as a current khat chewer or not.

10. The discussion art the authors tried to explain the higher odd of chewing khat at older age might be due to higher financial independence yet a few lines after, there is a contradictory explanation stating, unemployed individuals are more likely to chew khat. How can you explain these two assumptions?

Response: Older women are more likely to chew khat than younger women. The financial independence don’t necessarily come from being employed. Older women can get financial aid from their children and extended family members. This could also explain why more unemployed women are engaged in khat chewing than employed women.

11. Do the authors have any methodological recommendation for further study as the current study mentioned the limitations to consider factors such as the dosage of khat chewed per session and the duration of a woman's?

Response: The recommendation has been added.

12. Dear!!! you have to do again you report to make different from previous study on Prevalence and determinants of chewing khat among women in Ethiopia: data from Ethiopian Demographic and Health Survey 2016

Response: There has been a degree overlap with the mentioned study, however, the difference is much clearer when we look at the objective of our study. The objective of the study has been edited to give more emphasis to the points that make it different from the previous studies. Now the focus is on the association between khat chewing and malnutrition.

---

## [Editor Report · Decision Letter 1]

14 May 2024

Current khat (Cathaedulis F.) use among Ethiopian women and its association with anemia and underweight: A cross-sectional analysis from Ethiopian Demographic and Health Survey

PONE-D-24-01395R1

Dear Dr. Amaha,

We’re pleased to inform you that your manuscript has been judged scientifically suitable for publication and will be formally accepted for publication once it meets all outstanding technical requirements.

Kind regards,

Melese Shenkut Abebe, PhD

Academic Editor

PLOS ONE

Additional Editor Comments (optional):

Dear author

Thank you for addressing the reviewers and academic editor's comments. Now your manuscript is suitable for publication in PLOSE ONE. Note that your paper is subject to all outstanding content-related queries of PLOSE ONE. I wish you good luck.
---

## [Editor Report · Acceptance letter]

23 May 2024

PONE-D-24-01395R1 

PLOS ONE

Dear Dr. Amaha, 

I'm pleased to inform you that your manuscript has been deemed suitable for publication in PLOS ONE. Congratulations! Your manuscript is now being handed over to our production team.

Kind regards, 

on behalf of

Dr. Melese Shenkut Abebe 

Academic Editor

PLOS ONE